# Transluminal Pillars—Their Origin and Role in the Remodelling of the Zebrafish Caudal Vein Plexus

**DOI:** 10.3390/ijms242316703

**Published:** 2023-11-24

**Authors:** Helena Röss, Dea Aaldijk, Mykhailo Vladymyrov, Adolfo Odriozola, Valentin Djonov

**Affiliations:** 1Institute of Anatomy, University of Bern, 3012 Bern, Switzerland; helena.roess3@unibe.ch (H.R.); dea.aaldijk@unibe.ch (D.A.); adolfo.odriozola@unibe.ch (A.O.); 2Data Science Laboratory, University of Bern, 3012 Bern, Switzerland

**Keywords:** intussusceptive angiogenesis, caudal vein plexus, transluminal pillar, vessel remodelling

## Abstract

Intussusceptive pillars, regarded as a hallmark of intussusceptive angiogenesis, have been described in developing vasculature of many organs and organisms. The aim of this study was to resolve the question about pillar formation and their further maturation employing zebrafish caudal vein plexus (CVP). The CVP development was monitored by in vivo confocal microscopy in high spatio-temporal resolution using the transgenic zebrafish model *Fli1a:eGPF//Gata1:dsRed*. We tracked back the formation of pillars (diameter ≤ 4 µm) and intercapillary meshes (diameter > 4 µm) and analysed their morphology and behaviour. Transluminal pillars in the CVP arose via a combination of sprouting, lumen expansion, and/or the creation of intraluminal folds, and those mechanisms were not associated directly with blood flow. The follow-up of pillars indicated that one-third of them disappeared between 28 and 48 h post fertilisation (hpf), and of the remaining ones, only 1/17 changed their cross-section area by >50%. The majority of the bigger meshes (39/62) increased their cross-section area by >50%. Plexus simplification and the establishment of hierarchy were dominated by the dynamics of intercapillary meshes, which formed mainly via sprouting angiogenesis. These meshes were observed to grow, reshape, and merge with each other. Our observations suggested an alternative view on intussusceptive angiogenesis in the CVP.

## 1. Introduction

Intussusceptive angiogenesis (IA), also known as blood vessel splitting, is a complementary mechanism to sprouting angiogenesis (SA) that was proposed for the expansion and remodelling of vascular networks. The hallmarks of IA are thin transluminal pillars with a typical diameter of 1–2.5 µm. They are hypothesised to form via the simultaneous invagination of opposing vessel sides, perforation of the contact zone, and finally, invasion of perivascular cells and connective tissue fibres (reviewed in [1]). There is abundant evidence for the presence of these thin transluminal structures in developing embryonic vasculature of different species, such as in chicken chorioallantoic membrane (CAM) [2], [3,4], mice embryos [5], human embryos [6], developing frog [7] and most recently in zebrafish embryos [8,9].

Besides IA, two other main mechanisms of blood vessel formation have been described so far: sprouting angiogenesis and vascular mimicry. Sprouting angiogenesis can be stimulated by hypoxia in the tissue, leading to the expansion of the existing vessel network via the formation of sprouts, degradation of the extracellular matrix and consecutive migration of endothelial cells [10,11]. It is thus an invasive process with a high energy demand [12]. Vascular mimicry is a mechanism relevant in various, highly invasive tumour tissues, consisting of the transdifferentiation of tumour cells into endothelial-like cells [13,14].

IA, in contrast, serves for the growth and differentiation within an already existing vascular bed. It may have the following morphological and functional outcomes: (I) Intussusceptive microvascular growth (IMG): insertion of pillars may serve to expand the vascular bed within itself and increase its complexity. Since it requires only a small proliferation rate, this mechanism enables the growth of a plexus with low costs of energy. IMG has been documented in numerous tissues and animal models and is therefore thought to be universal. (II) Intussusceptive arborisation (IAR): pillar arrays may merge and split small arteries and veins to establish a hierarchy in the vascular network. (III) Intussusceptive branching remodelling (IBR): pillars may also enable optimisation of the branching geometry by changing the branching angle towards the ideal of a constant level of shear stress with minimal power usage (“Murray’s Law”). Hereby, unlike in IMG and IAR, the number of vessels remains the same or can be reduced by pruning (reviewed in [15]). In line with the findings of IBR, the insertion of transluminal pillars was shown to be flow-dependent and triggered by a local drop in shear stress. This has been demonstrated in vivo [4], in vitro [16] and in computational models [17,18] (reviewed in [19]). IA does not require a remodelling of the surrounding matrix or endothelial cell (EC) proliferation and migration, but the ECs rather increase their size and flatten [20]; it is thus considered to be a more energy-efficient process [20]. Therefore, it is not surprising that the intussusceptive pillars have been frequently observed in various types of tumours and they may also be a therapeutic target in tumour treatment when anti-VEGF drugs fail [21], reviewed in [22].

Different mechanisms of pillar formation have been proposed: intraluminal capillary wall fusion [23], capillary wall folding [24], and inverse sprouting [25]. The description of these mechanisms is based on different developmental stages that have been captured on serial semithin or transmission electron microscopy (TEM) images, vascular casts, and with light microscopy images. The currently preferred “intraluminal capillary wall fusion” model proposes four stages of pillar formation: (I) opposite sides of the endothelium make intraluminal processes that contact each other; (II) the endothelial intercellular junctions reorganise, and the contact zone is perforated; (III) cytoplasmic extensions of myofibroblasts, pericytes, and interstitial fibres infiltrate the pillar, forming its core; and (IV) merge to create a thin cylindrical pillar, which eventually grows in diameter as it becomes penetrated by perivascular cells and extracellular matrix [26]. The alternative “capillary wall folding” mechanism proposes pillar formation by a perforation of a one-sided intraluminal fold [24]. In more detail, first, an intraluminal fold consisting of collagen fibrils and perivascular cell extensions is formed. Next, this fold becomes thinner and thinner until it contains only a bundle of collagen fibrils coated by a single endothelial cell layer (pre-pillar). Finally, the fold is perforated, causing the separation of the pillar. Yet another mechanism termed “inverse sprouting” was proposed much later [25]. According to it, first, a thin transluminal endothelial bridge is formed (how exactly remains open). The endothelial cell then pulls collagen bundles inside this bridge to stabilise it until it stretches throughout the pillar.

Although there are over 200 publications documenting the presence of transluminal pillars in different vessel beds, their development and subsequent growth have, to our knowledge, never been documented in vivo in 3D. Recently, transluminal pillars have also been observed in the caudal vein plexus (CVP) of zebrafish embryos [8]. The transparency of the zebrafish embryonic tissue, the flat morphology, the small dimensions of the CVP (ca. 700 × 200 × 50 µm) and the feasibility of in vivo imaging [27] make the CVP an ideal model to address those questions.

The CVP is a transient venous network in the tail of zebrafish, located caudally of the yolk sack extension. It starts forming at 24 h post fertilisation (hpf) when the heart starts beating, and blood creates a simple loop between the dorsal aorta and a caudal vein. The development of the plexus has been described as follows: the starting point is marked by sprouting from the posterior cardinal vein and fusion of neighbouring tip cells. Sprouting from the venous segments in a caudal-ventral direction and rapid anastomosis leads to the formation of a primitive plexus by 32–36 hpf [8,28]. Then, the plexus continues to expand, and hierarchy begins to be established. By 48 hpf, the majority of the circulating blood cells are restricted to a vascular loop formed by the dorsal aorta and the most ventral branch of the venous plexus (the caudal vein). The process of gradual elimination of the redundant loops and parallel branches of the plexus further continues until 3 days post fertilisation (dpf), and it retracts in a dorso–ventral direction, also due to the tail growing in length. The redundant branches from the central part of the plexus do not fully retract, as observed in pruning since they serve as a haematopoietic niche [29]. Between 3 and 5 dpf, some of these connections are observed to drain the blood from the intersegmental vessels to the caudal vein. At 5–7 dpf, the plexus has been simplified to one major loop between the dorsal aorta and the caudal vein [30].

Another question that has not been answered so far concerns the transition from sprouting to intussusception. As documented by various methods in different developing vascular beds, the high number of sprouts during early plexus formation indicates that this phase is dominated by SA (reviewed in [31]). In later stages of the development, the sprout density decreases, and the number and diameter of thin intraluminal pillars gradually increase, which points to the participation of IA in subsequent plexus remodelling. There is, however, no gold standard for the quantification of the involvement of SA versus IA. The semi-quantitative approaches include pillar count versus sprout count [8,32], pillar count versus hollow (big meshes) count [33], or the use of endothelial cell proliferation as an indicator of SA and the presence of pillars as a marker of IA [34]. Other studies quantify IA based on the pillar count. However, “pillars” may be “small holes” on corrosion casts without specification of the diameter [35], structures of several µm in diameter seen on wild field acquisitions [36,37], or a circular intraluminal structure [38], a fold [39], an intraluminal contact intermediate of a “pillar” on a histology section [40], or a cross-section acquired by TEM [41]. Many studies use a combination of several methods to describe the wave of IA [42,43,44], but the classification criteria for intussusceptive pillars, when specified at all, are difficult to compare.

Little is known also about the molecular mechanisms of IA, and there are sometimes contradictory results on the involvement of major angiogenic pathways—depending on the model and the methodology used [20]. The most frequently mentioned regulators of intussusception are hemodynamic forces, the VEGF pathway, and EphrinB2/EphR4 signalling.

Blood flow velocity and shear stress have been reported to affect IA. In computer models, the appearance of pillars in wild-type models, as well as vessel splitting, could be predicted [8,45]. The absence of blood flow in disease models may favour the formation of pillars that, however, fail to trigger vessel splitting (abortive intussusception) [46].

The VEGF/VEGFR2 signalling pathway is the major pro-angiogenic pathway that stimulates endothelial cell sprouting. VEGFR2 is a tyrosine kinase receptor expressed on endothelial cells; its ligand VEGF is expressed under hypoxia [47]. ECs that are exposed to higher VEGF concentrations behave as leading tip cells that possess numerous filopodia, sense the VEGF, and migrate towards the hypoxic regions. The non-tip cells receiving lower VEGF concentration, the so-called stalk cells, merely follow the migrating front and form no or only a few filopodia. Vascular meshes form via the interaction of two tip cells and anastomosis—a process that has been documented in much detail already [48,49]. The filopodia, actin-based membrane protrusions, are not required for tip cell migration, but they facilitate anastomosis [50]. Inhibition of the VEGF pathway by engineered antibodies or low molecular weight inhibitors is the basis of antiangiogenic therapy [43]. Some studies reported that VEGF favoured IA [20,51,52], others the opposite [20,53,54,55].

EphrinB2/EphR4 signalling plays a central role in the remodelling of vascular beds, venous sprouting, splitting angiogenesis [56], and especially in arteriovenous specification [57], which is reviewed in [58]. Higher expression of the receptor tyrosine kinase EphB4 is linked with venous specification, while its ligand EphrinB2 is mostly arterial-specific. The receptor–ligand interactions can facilitate endothelial cell migration, which is critical in both embryonic angiogenesis, as well as in diverse pathologies. Mutations in the EphB4/RASA1/MAPK pathway cause arteriovenous anomalies [59]. Two studies mentioned its role in promoting intussusceptive angiogenesis [54,56].

To summarise, there is broad evidence for the presence of intraluminal pillars in the developing vascular beds of different species during vessel reorganisation. Nonetheless, little is known about the mechanisms of the formation and subsequent growth of pillars and intercapillary meshes. Zebrafish embryos present a valuable and well-documented model for angiogenesis research with the possibility of in vivo imaging (reviewed in [60]). We chose the zebrafish CVP to document the mechanisms of pillar formation. The zebrafish lines *Tg(fli1a :eGFP)y7* [61] and *Tg(gata1: DsRed)sd2* [62] provide excellent models to study blood vessel development since they express a green fluorescent protein (GFP) within the endothelial cells and red fluorescence protein (DsRed) in the erythrocytes, respectively. Since shear stress has been shown to be an important factor in the formation and maturation of blood vessels, zebrafish of the *sih* line with a silent heart (no heartbeat, bearing a mutation in *tnnt2a*, a gene encoding cardiac troponin T) [63] were used to study the caudal vein plexus without the influence of shear stress. The aim of this study was to gain insight into the role of pillars in plexus expansion and remodelling in high spatio-temporal resolution.

## 2. Results

### 2.1. Appearance, Morphology, and Distribution of Pillars in the Developing CVP

Previous data suggest that the CVP develops between 30 and 42 hpf, mainly via IA in proximal perfused regions and by continuing SA in distal non-perfused regions [8]. In line with that, we observed numerous transluminal pillars present in the CVP during that time (Figure 1a,c).

At 48 hpf, the caudal vein plexus consisted of a single artery and a venous plexus, which originally had a honeycomb-like pattern (Figure 1a (i)). Short vessel segments encircled the intercapillary meshes (rectangles, Figure 1a (ii)), and further complexity was added to the plexus by thin intraluminal pillars (circles, Figure 1a (ii)). Most pillars were found in the venous part of the CVP, along the plexus border rather than in the centre, and these pillars were perpendicular to the plane of blood flow (Figure 1a (iii,iv) and examples in Figure 2 and Figure 3). Occasionally, a pillar was present in the most caudal part of the dorsal artery, close to where it made a loop towards the caudal vein. Such pillars had a flat angle to the body symmetry plane and were longer than the vessel diameter (Figure 4).

In in vivo confocal scans, meshes had a sand-clock or biconcave profile (Figure 1a (v to viii), while pillars appeared as straight cylindrical structures (Figure 1a (ix to xii)) and, furthermore, showed typical morphology and content on TEM (see Appendix A). The average pillar diameter was 1.7 ± 0.7 µm and they had an average height of 11 ± 4.6 µm (Figure 1b, *n* = 67, measured at 48 hpf).

The pillar count per plexus reached its maximum at around 36–48 hpf. From 3 dpf (72 hpf) onwards, there was a highly significant reduction in the pillar number. Note the very high biological variability of pillar numbers and the persistence of many pillars even until 4 dpf (96 hpf) (Figure 1c) and likely onwards, although plexus remodelling had been mostly completed by that time.

### 2.2. Mechanisms of Pillar Formation

To investigate how these pillars formed, we imaged the developing CVP using time-lapse confocal microscopy from the end of the sprouting phase to the whole intussusceptive phase and beyond (between 30 and 48 hpf). Then, we tracked 74 pillars (defined by their morphology at 48 hpf) up to 16 h back in time. What we could observe were three main processes behind the pillar formation: (1) interaction of filopodia and tiny sprouts with each other and (2) lumen formation or (3) lumen expansion.

The common precursor of pillar formation was an intraluminal fold, whose perfusion led to the creation of a pillar. If it formed from an assembly of endothelial cells during lumen formation, we spoke of lumen expansion (LE, Figure 2, Appendix A) as the major force. If the intraluminal fold arose by merging of tiny sprouts from a lumenised vessel segment, the predominant mechanism was sprouting angiogenesis (SA, magenta arrows, Figure 3, Appendix A, Appendix A). And finally, if the intraluminal fold arose from a vessel without the apparent participation of sprouts, we borrowed the term capillary wall folding (CWF, Figure 4, see also Appendix A, Appendix A) (the driving force seemed to be the growth of the vessel diameter but a barrier outside the vessel stood in the way) because it resembled the mechanism proposed by Patan et al. [24].

Most pillars were formed by a combination of SA and LE (yellow arrows, Figure 3
Appendix A) or by a mixture between SA and CWF, but sometimes one of them seemed to be dominating over the others, as seen in the selected examples. These processes were reversible, and sometimes the intraluminal fold was smoothened again without a pillar being formed. Even an already-formed pillar could retract back to a capillary wall fold and eventually fully disappear.

We also observed thin “pillars” reminiscent of endothelial bridges documented by Paku et al. [25] (Figure 2 and Figure 4). Endothelial bridges, as the name suggests, consist only of endothelial cell connections spanning through the lumen, with no visible “core”, and they emerge from the luminal side of endothelial cells. However, these fragile connections were very unstable, became thinner and broke within an hour or two. Another example of the formation of a pillar by SA and LE and its subsequent rupture can be seen in Appendix A (the formation by SA and LE, as well as the behaviour of pillars also in Appendix A).

**Figure 2 ijms-24-16703-f002:**
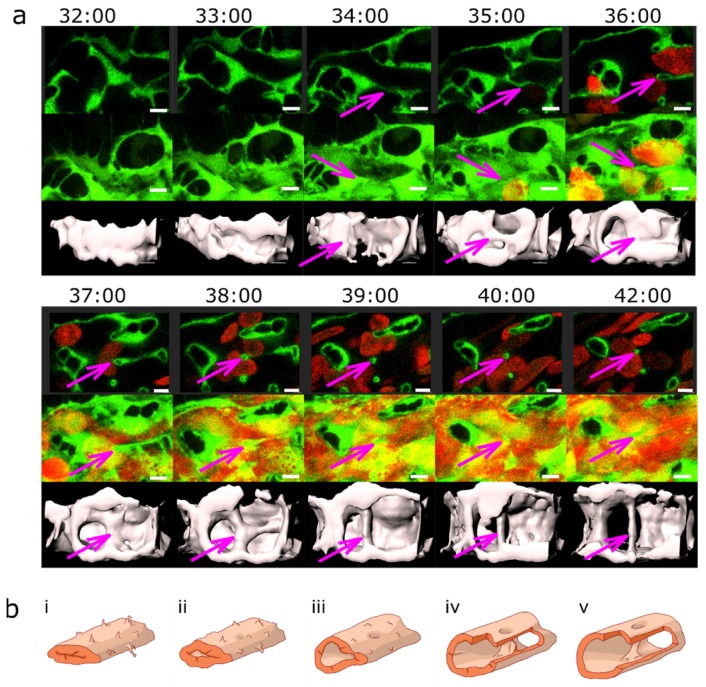
Lumen expansion and pillar formation. (**a**) Confocal images showing the formation of a pillar by lumen expansion. The pillar is marked with a magenta arrow. The example shows the high plasticity of the developing endothelial network. The pillar starts being visible in the 3D reconstruction at time point 35:00 and in the single slice at time point 36:00. As the pillar forms, weak connections to the endothelial walls (endothelial bridges) may be apparent, as seen in the 3D reconstruction at time point 38:00 but they become thinner and break soon (at time point 39:00). First row: one confocal section of 1 µm; endothelium in green (eGFP); erythrocytes in red (dsRed). Second row: maximum intensity projection. Third row: 3D reconstruction showing surface models of the endothelium in grey (based on the GFP channel). Time points (hh:mm post fertilisation) are valid for each column. Scale bars: 5 µm. (**b**) Illustration of the proposed mechanism. The core of the pillar is supposedly formed in the sprouting phase (**i**,**ii**, visible as a shadow on (**iii**)) but the pillar as such form as the vascular lumen forms (**ii**–**iv** and finally (**v**)). Endothelial cell borders are not shown, the cellular extensions represent filopodia.

**Figure 3 ijms-24-16703-f003:**
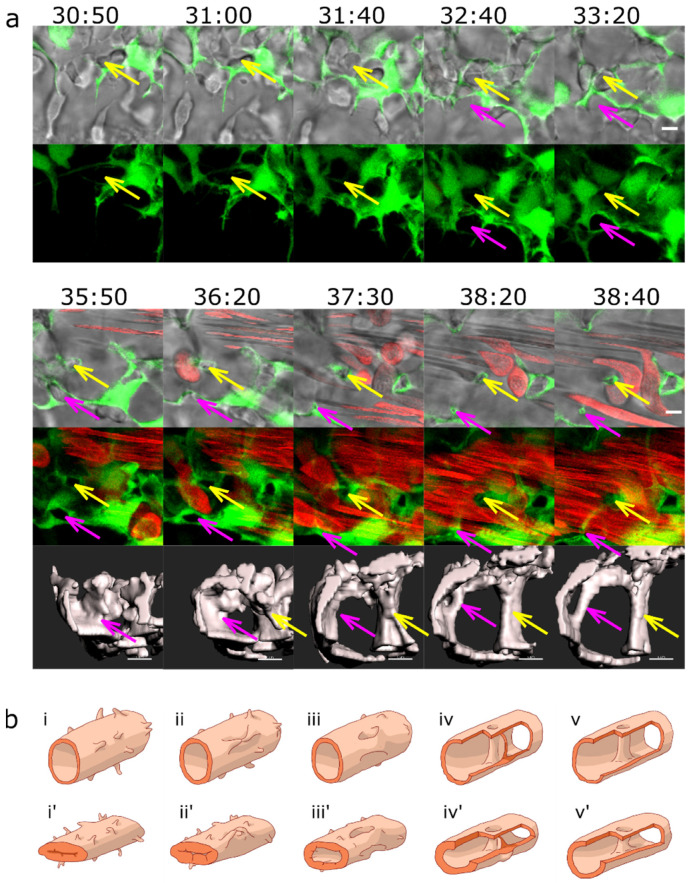
Pillar formation via fusion of sprouts. (**a**) Confocal images showing the formation of two pillars (indicated by coloured arrows) that could be tracked back to sprouting angiogenesis. First, endothelial cells form highly dynamic filopodia that make transient contacts (at time points 30:50 and 31:00). Second, they form small loops that are not perfused first, visible at time point 32:40 (yellow arrow) and 33:20 (magenta arrow). This stage corresponds to the “intraluminal fold”, except that the lumen is very narrow and not yet well defined. Third, vessel perfusion starts: the lumen expands, and the pillar detaches from the vessel wall. This can be seen at the time point 37:30–38:40 (magenta arrow). The pillars grow in z-axis (usually perpendicular to the plane of the CVP), and the pillar diameter often shrinks. The structure is morphologically visible as a pillar only now. Endothelium in green (eGFP); erythrocytes in red (dsRed). First row: single slice of 1 µm with bright field channel added. Second row: maximum intensity projection. Third row: 3D reconstruction in grey. All confocal images are in the same magnification. Time points (hh:mm post fertilisation) are valid for every three (two) representations below. Scale bar: 5 µm. (**b**) Illustration of the proposed mechanisms; (**i**–**v**): sprouting (corresponding to the pillar marked by the magenta arrow); (**i’**–**v’**): combination of SA and LE (corresponding to the pillar marked by the yellow arrow).

**Figure 4 ijms-24-16703-f004:**
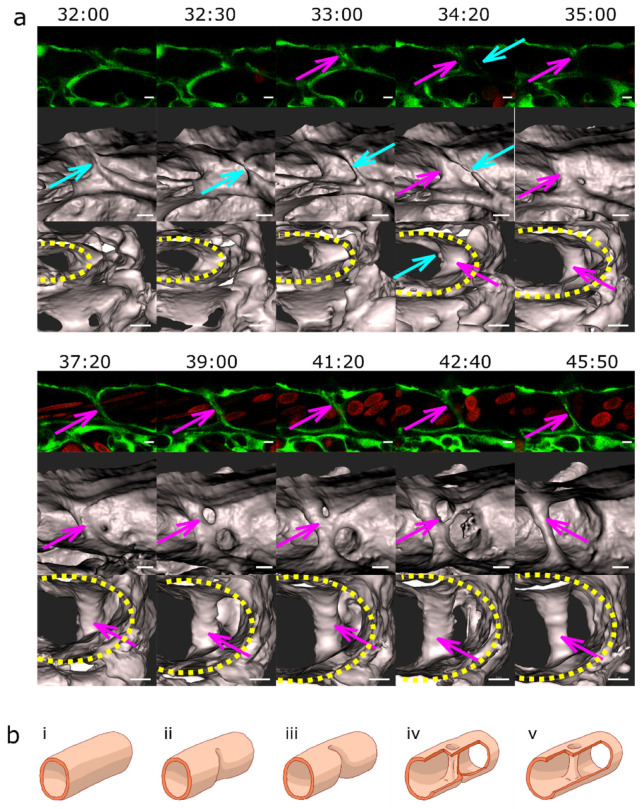
Pillar formation by capillary wall folding. (**a**) Development of one of the strong transversal pillars that are frequently observed in the caudal part of the dorsal aorta (DA). The pillar starts to form as an intraluminal fold (time point 34:20; magenta arrow) and grows as the lumen of the DA expands. Briefly, another thin pillar (cyan arrows) or endothelial bridge forms via the same mechanism (time point 32:00–33:00) but becomes thinner and ruptures soon afterwards (34:20). First row: one confocal section of 1 µm; endothelium in green (eGFP); erythrocytes in red (dsRed). Second row: maximum intensity projection of the same region. Third row: surface models of the endothelium in 3D (based on the GFP channel) from two different perspectives; the yellow dotted line shows the lumen of the dorsal aorta. Time points (hh:mm post fertilisation) are valid for each column. Scale bars: 5 µm. (**b**) Illustration of the proposed mechanism based on the 3D reconstructions above. Formation of the intraluminal fold (**i**–**iii**), its thinning (**iv**) and its perforation leads to the formation of pillar (**v**).

### 2.3. Blood Flow and Pillar Formation

According to the literature, major stimuli for pillar formation (whether by intraluminal invagination or by the alternative mechanism) are steep changes in blood flow and shear stress. We analysed Fli:eGFP//*sih* embryos completely lacking blood flow and thus with no local shear stress differences (Figure 5a i and ii, Appendix A). Before the onset of circulation (24–26 hpf), the vasculature of the silent heart (*sih*) embryos resembled that of wild-type (wt) siblings (in agreement with [64]), but at 48 hpf, there was no vascular network formed. The venous part was inflated and filled with blood cells, and there were numerous thin pillars in the lumen. The main mechanism of pillar formation was a fusion of sprouts, alone or in combination with lumen expansion, which are the mechanisms seen in wt embryos, too (Figure 5b). So, even though pillars were present in *sih* embryos, they did not contribute to the development of a functional vessel network. The total number of pillars in *sih* was higher than in wt (count: 13, 22, 21, 19, average 18 ± 4, *p* = 0. 1735 × 10^−7^). They were somewhat thinner (1.59 ± 0.54 µm) and longer (14.2 ± 3.1 µm, *n* = 60). What was much more striking was that there were almost no meshes at 2 dpf, and the meshes present were notably smaller than in wt (compare the absolute area of pillars and meshes in *sih* versus wt in Appendix A). Furthermore, the pillars were not observed to rupture, but sometimes they merge with other nearby pillars. This demonstrated that pillar formation by fusion of sprouts does not require blood flow, which is in agreement with the fact that sprouts can arise from unperfused vessels.

### 2.4. Further Fate of Pillars and Intercapillary Meshes

Previous studies linked pillars to vascular splitting: small pillars are thought to expand and evolve into meshes, and pillars or meshes in an array may fuse with each other to form a longitudinal slit. We investigated in vivo whether the pillar or mesh cross-section areas increased in time and whether serried pillars or meshes merged with each other.

The cross-section area of meshes and pillars was measured at time points 28–32 hpf and again at 44–48 hpf. We calculated the absolute area change and plotted it against the cross-section area at the first time point (Figure 6a). There was a positive correlation between pillar/mesh size and its growth (black rectangles) and also between its size and its diminishment (grey rectangles). The bigger the structure, the bigger the absolute change in its area over time.

We further plotted the count of pillars/meshes against the relative area change (Figure 6b). Only 1 out of 17 pillars (or very small meshes, the area below 12.5 µm^2^, corresponding to a diameter of 4 µm or less, *n* = 17) changed its cross-section area by more than +50%. The average relative cross-section area change was −16 ± 68% (absolute average change +1.1 ± 9.9 µm^2^). In contrast, the majority of meshes (area over 12.5 µm^2^, *n* = 62) increased their area by more than +50% over time (39 out of 62 meshes). The average relative difference in the cross-section area of meshes between the two time points was +95 ± 110% (absolute average change +112 ± 150 µm^2^). Even among meshes of similar size at 1.25 dpf, there were some that grew less, some that grew more, and some that decreased their cross-section area, which led to a relatively high SD (see Appendix A with a plot of the absolute areas). The behaviour of very small meshes and pillars was more uniform. In *sih* embryos, there were very few meshes left at 2 dpf, and their area was very small in comparison to wt (also Appendix A).

The increase in mesh area was flow-dependent: all the meshes and pillars in *sih* embryos decreased their cross-section area over time (Figure 6c,d).

### 2.5. Merging of Adjacent Pillars/Meshes

Of 78 pillars that were observed between 28 and 32 hpf, two-thirds (*n* = 52) remained present even at 48 hpf. One-third of the pillars (*n* = 26) disappeared between 32 and 48 hpf, either by merging with a vessel wall or a nearby mesh (*n* = 17, more frequently with an outer vessel branch than with an inner mesh) (Figure 7c,d), see also Appendix A, Appendix A) by a process that resembled reversed ”capillary wall folding” and led to the disappearance of the barrier to blood flow. Since there was no observable increase in the mesh diameter after it had fused with the pillar, it appeared that this process did not contribute to vessel splitting.

A further mechanism of pillar disappearance was thinning and subsequent rupture (*n* = 9, Figure 7, cyan arrow). It began with elongation of the pillar, rupture of its core, and then rapid thinning and rupture of the pillar envelope, too. In contrast to our expectation, we did not observe the appearance of new serried pillars at the branching points and expected vessel splitting by merging of serried pillars (or small meshes) often did not take place (Figure 7d).

Plexus remodelling was driven by the behaviour of meshes, namely their merging (Figure 7a) or, less frequently, their fragmentation (Figure 7b) and their growth (as described above and Figure 7b). Most of the meshes tracked back from 48 hpf were present already at 28 hpf; the new ones formed usually by sprouting angiogenesis at the growing front of the plexus or less frequently by fragmentation of a mesh.

The first step in the merging of meshes seemed to be the reduction in the flow in the connecting branch between the two meshes. As the flow ceased and eventually stopped, the lumen of the connecting branch closed like a camera aperture (as if the pressure that held it expanded disappeared and the tube now relaxed) (Figure 7a). The opening in the lumen closed and eventually disappeared, and then the endothelial cells that had formed this connection retracted away, similarly to the process named “vascular pruning”.

Splitting of an existing mesh was occasionally seen in very narrow ones, where the two opposite branches eventually touched each other and became connected (Figure 7b); also compare the dynamics of meshes in Appendix A. Splitting and merging were often reversible, as documented in the selected example.

### 2.6. Plexus Simplification and Establishment of Hierarchy

The major events in plexus remodelling were arteriovenous or venous splitting (Figure 8, Appendix A) that occurred by the fusion of neighbouring meshes. As the plexus expanded caudally, numerous meshes formed in the distal part. The onset of circulation is around 24–26 hpf and around 28–32 hpf; there were numerous connections between the dorsal artery and the venous plexus. The less used ones located most cranially began to close first. Simultaneously, the plexus continued to grow, the lumen gradually opened in the caudal branches, and meshes located there expanded and fused with each other, too. Around 48 hpf, the dorsal aorta is usually drained to the venous plexus via one or a few branches.

The closure of the redundant connections between DA and veins went along with a reduction in flow through them, but since the arteriovenous splitting took part in *sih* embryos too, we suggest that this happened rather due to blood flow differences or a gradient than due to the absolute values.

Venous splitting was not so well predictable but followed the same principle. It took place at the onset of perfusion in the peripheral parts of the plexus (Figure 8c) or as flow conditions changed (Figure 8b and Appendix A). However, although the venous flow was almost exclusively restricted to the most ventral branch of the venous plexus, most meshes in the central part of the plexus did not merge with each other by 2.5–3 dpf. Merging of the meshes was not solely responsible for the establishment of a hierarchy between small and big veins.

Redirection of the flow through the most ventral vessel (the caudal vein) occurred via lumen collapse or even closure of the segments forming the central part of the plexus. As the blood flow through the central part of the plexus dropped between 2 and 3 dpf, the lumens in the interconnecting segments gradually shrank and eventually collapsed (Figure 8d). The lumen closure was not followed by endothelial cell retraction, as would be the case in pruning.

Inhibition of the EphrinB2/EphB4 signalling pathway with 1 µM BHG712 resulted in a distinct phenotype (Figure 9): the dorsal aorta was only to two-thirds separated from the venous plexus (compared with normal development in Appendix A). The distal part of the plexus was lumenised but not perfused, and it contained numerous pillars and small meshes. Like in *sih* (and wt controls), they formed in the sprouting phase and not later via the intraluminal capillary wall folding. The average pillar count in BHG712-treated embryos was significantly increased (average 21.5 ± 3.8 compared to 5.1 ± 3.3 in wt, *p* < 10 × 10^6^), and most of them (ca 2/3) were localised in the nonperfused region.

In our pilot study, inhibition of the VEGFR2 pathway did not (Appendix A and Figure 9) significantly affect the pillar count and arteriovenous splitting, although we observed diverse plexus anomalies (Appendix A). The only other inhibitor associated with significantly increased pillar count (10.1 ± 5.5, *p* = 0.03) was Latrunculin B (LatB), an inhibitor of filopodia formation. The CVP of LatB-treated embryos also contained lumenised segments without blood flow, indicating (together with the *sih* data) the common predictive factor for pillar count.

## 3. Discussion

The concept of intussusceptive angiogenesis was first presented in 1986. It describes the formation of pillars within the vessel lumen and their fusion, leading to vessel splitting and remodelling [65]. Since then, a large body of evidence has been collected on this phenomenon [20]. The morphology of intraluminal pillars and their existence in different vessel beds has been well documented, but the dynamic aspects of this process, namely how the pillars form in vivo and how they facilitate vessel splitting, remained largely unexplained. The availability of suitable transgenic models and the improvements in confocal microscopy in recent years made such studies possible. We used the zebrafish caudal vein plexus model that was established in our laboratory by Karthik et al. [8] and documented the formation of the transluminal pillars and their contribution to the plexus development in high spatio-temporal resolution.

Most pillars were present in the venous part of the CVP, and they were orientated perpendicular to the symmetry plane of the embryo, but one or a few pillars were usually present in the distal part of the dorsal aorta (oriented rather diagonally). Pillars were most numerous between 32 and 48 dpf, with very high interindividual heterogeneity. There was a significant reduction in the pillar number from 3 dpf onwards, although some of the pillars persisted until at least 4 dpf. These data correlate with previous findings in our group [8]. We measured an average pillar diameter of 1.7 ± 0.7 µm and an average length of 11 ± 4.5 µm, and their ultrastructure corresponded to the intussusceptive pillars documented before (see Appendix A with TEM/3View images of pillars).

A problem in comparing our results with other studies is the paucity of quantitative data on pillar dimensions and the inconsistency of literature regarding the definition of a pillar. Most studies defined a pillar as a cylindrical structure with a diameter of 1–2.5 µm seen on SEM or on serial TEM sections; some extended the cut-off to up to 4 µm [66] or did not specify a cut-off at all. Föhst et al. developed an automated algorithm that classified all transluminal columns of a given diameter (they decided on 1–5 µm) [67], and they reported a highly asymmetric distribution of pillar diameter and volumes, with most pillars being between 1 and 3 µm in diameter and having a volume of less than 40 µm^3^ (our average measurements of diameter and length would correspond to a volume of 25 µm^3^). To see if the (most cited) cut-off at 2.5 µm was appropriate, we included in our quantifications (diameter and height, Figure 2, and area change, Figure 7) structures up to 4 µm (or up to 12 µm^2^ in cross-section). Most pillars were indeed in the range of 1–2.5 µm and had the typical cylindrical shape.

Among the 74 pillars that were tracked back to their formation, we observed three processes involved in pillar formation: sprouting (formation of filopodia that “embrace” the future pillar core), lumen formation/expansion, and formation of a fold in an already lumenised vessel (capillary wall folding). This does not mean that there were three distinct mechanisms; it was rather a highly heterogeneous process where these three events participated to a certain degree, possibly depending on how mature the vasculature was at the time of pillar formation. Pillar formation by capillary wall folding has already been proposed before by Patan et al. [24] based on TEM data. We could identify vertical tissue folds and intermediates that they termed “pre-pillars” as immediate predecessors of the pillars that we observed by confocal microscopy. According to Patan et al., the pre-pillars formed in already lumenised vessels after the sprouting had ceased. Our data indicated that they may also appear earlier, during lumen formation, and have their origin in sprouting angiogenesis. We also observed thin intraluminal connections reminiscent of endothelial bridges described by Paku et al. [25], but their nature was more transient, and they seemed to have no particular function. Contrary to our expectations, none of the observed thin transluminal pillars formed via the intraluminal invagination and perfusion of the contact zone.

Pillar formation is thought to occur preferably at branching points of vessels, triggered by a local drop of shear stress [18]. We used *sih* embryos (bearing a mutation in cardiac troponin and thus totally lacking blood flow) to test whether the pillar formation was flow-dependent. Our results showed an impairment of mesh formation as reported previously by Xie et al. [64], but pillars were present nonetheless, and their number was increased in comparison to wt embryos with resembling mechanisms of formation. The insertion of pillars in the CVP of *sih*-embryos was therefore contrary to previous findings [4,8], not driven by local differences in blood flow or shear stress. An important remark here is that the computer models of flow-dependent pillar insertion were built on the intraluminal capillary wall fusion mechanism, which we did not confirm in the developing CVP. Yet another point might be that the in silico calculations [18] and in vivo data showing the dependence of pillar formation on the blood flow referred to structures of 5–10 µm as “pillars” [8]. The formation of structures of this diameter (termed “meshes” in our work) was, according to our data, indeed impaired under no-flow conditions. The higher number of pillars in *sih* was likely caused by two factors: firstly, some of the collagen bundles that would normally be inside a mesh in wt were present as a pillar because the vascular lumen was inflated instead of forming a regular plexus. Secondly, due to the absence of blood flow as a major force responsible for the pillar rupture, most of the formed pillars persisted. A similar phenotype to *sih* with segmental dilatation of the caudal vein and pillars resembling intussusceptive pillars has also been observed in zebrafish embryos with mosaic inactivation of *ccm2* using morpholino oligos [46]. Similarly, they failed to fuse and therefore split pre-existing vessels. The authors described this process as abortive intussusceptive angiogenesis, identified it as a consequence of aberrant flow signalling, and pointed out the similarities to human cerebral cavernous malformations. The authors did not report similar pillars in the control embryo. Among our controls, there were examples with no pillars, too (see Figure 1, 2/29 embryos at 48 hpf).

The concept of intussusception postulates that intercapillary meshes evolve from transluminal pillars that increase their diameter as they become penetrated by perivascular cells (e.g., myofibroblasts or pericytes [1,26,68]). To examine this hypothesis, we tracked back meshes in time. Between 28 and 48 hpf, there were some newly formed meshes (by SA) or pillars, and there were some cases of mesh merging or splitting, but the majority of structures only changed their size. While the majority of meshes grew (39/62 grew by more than +50%), the majority of pillars (or very small meshes: the cross-sectional area below 12 µm^2^ or below 4 µm in diameter) did not change much or shrank (only 1/17 changed their area by more than +50%, namely by +209%). This growing structure originally had a diameter of 3.5 µm; without information about the ultrastructure, we cannot tell with certainty whether this was a true case of pillar growth or whether, which we tend to believe, this was a tiny mesh already from the beginning.

The intercapillary meshes were thus, contrary to our expectations, not a later stage of intussusceptive pillars but formed by SA [28,69]. These meshes were highly dynamic and effectively shaped the vascular bed in the later stages of plexus remodelling and simplification. Pillars usually persisted in the lumen for several hours without any change in diameter or shape, and many eventually disappeared by rupture or fused back with the vessel wall (“capillary wall folding” reversed). The average lifetime of the observed pillars cannot be calculated since our 14–16 h long scans contained only a few cases of the complete life of a pillar. A possible reason for the observed lack of growth of slender pillars might be a strong interaction between endothelial cell receptors and extracellular matrix (ECM) constituents of the pillar core [70] that simply do not allow infiltration of pillars by perivascular cells. In contrast, small meshes might have a core consisting of diverse ECM components and cell extensions that are less dense and thus more plastic. In TEM images, it can be seen that pillars contain merely ECM, whereas, in meshes, there are endothelial cells visible (Appendix A).

Our data also remind us of the recently proposed piecemeal mechanism that integrates IA and SA [71]. They described it as a combination of sprouting and intussusception angiogenesis, where sprouting stands at the beginning of the process and the appearance of pillars/papillae shows intussusception to be part of the following process.

So, how can we integrate the findings of our study into the broader picture of earlier research in this field? The involvement of SA during the early development of the CVP [28], followed by the appearance of pillars at 36–48 dpf in our study, recapitulates the previous findings well. The dynamics of pillar count per CVP observed in our study are in line with the data of Karthik et al. [8].

However, our data on the mechanism of pillar formation, the behaviour of pillars and the process of vascular splitting contradict the long-established concept of intussusceptive angiogenesis. Firstly, pillars did not form by the capillary wall fusion mechanism or by an active intraluminal capillary wall folding. They rather appeared as “by-products” of the sprouting phase. Secondly, of over sixty pillars <2.5 µm in diameter that have been tracked in 3D for several hours, none increased its diameter, and none facilitated vessel splitting [8]. And thirdly, vessel splitting was dominated by the behaviour of meshes that arose by SA.

On the other hand, the initial increase in plexus complexity (pillar and mesh formation) resembled the IMG described in earlier studies about IA [20]. In line with the concept of IMG, the increase in vessel surface area did not appear to be attributable to endothelial cell proliferation but rather to the migration and flattening of endothelial cells (qualitative observations). Furthermore, the later CVP development followed the pattern described as IAR. The growth and merging of intercapillary spaces served to split vessels, and changes in lumen diameter further established the hierarchy. The third possible outcome of intussusception, namely pillar formation being involved in branching point remodelling as proposed in IBR, has not been observed in CVP. We observed pillar disappearance (by rupture or by fusion with the vessel wall), not their formation, in response to changed flow conditions.

The question is also if the formation of pillars and meshes is regulated by the same molecular pathways. In pilot experiments conducted in our institute, inhibition of the VEGFR2 pathway with sorafenib or cediranib did not result in a significantly different pillar count at 48 hpf in comparison to wt (Appendix A). This is in line with previous experiments with four other VEGFR2 inhibitors in an in vitro microfluidic model of IA [52]. Inhibition of filopodia formation by LatB resulted in an interesting phenotype with longer segments of ventral vessels, forming blind-ending tubes or meshes that were larger than in wt. This phenotype recapitulated the previous findings of Phng et al. well [50]. The number of pillars after LatB treatment was slightly higher than in controls (10 ± 6), which might be due to some unperfused parts of the plexus (like in the *sih* model), where pillars were not subjected to the same level of mechanical stress as pillars present in the circulation. The altered plexus morphology was, however, a product of impaired sprouting angiogenesis, not of altered plexus remodelling via splitting.

Treatment with the EphrinB2/EphB4 signalling inhibitor BHG712 resulted in a phenotype with a significantly increased number of pillars (21.5 ± 3.8 compared to 5.1 ± 3.3 in controls) and, in line with previous studies, impaired arteriovenous splitting [57]. The caudal part of the CVP (estimated one-quarter to one-third) was not perfused at 48 hpf because of the failure of arteriovenous splitting. A similar phenotype has recently been documented in 30 hpf embryos with a loss of Rasa1, a gene linked with arteriovenous malformations [72], which is not surprising given the known interaction between the Ras and Ephrin signalling pathways [73]. We also observed numerous smaller meshes and pillars in the nonperfused part (see Figure 9). Instead of naming EphB4 a positive regulator of pillar formation, the increased pillar count could likely be linked to the partial lack of perfusion and thus reduced mechanical forces on the pillar. This interpretation is in line with our *sih* data, as well as with the observation of other nonperfused regions of the CVP (such as in blind-ending segments after LatB treatment). Ultra-low shear stress has also been identified as a pillar-predicting factor by Arpino et al., which corroborates this explanation [52].

What mechanisms would reduce pillar count? Increased blood flow velocity and shear stress are well-known factors [8,18]. Furthermore, treatments that destabilise the collagen could impair pillar formation since collagen fibres stabilise the pillar core [8]. But because collagen is a ubiquitous component of the ECM, such embryos would likely have a complex phenotype. Our pilot data with VEGFR2 inhibitors did not fully reject the involvement of this pathway in both mesh and pillar formation—the lower pillar count in those embryos could be caused by decreased pillar formation. Alternatively, higher shear stress/increased blood flow would be linked with lower pillar count due to increased elimination of existing ones.

Numerous studies have already attempted to identify the molecular mechanisms behind pillar formation and intussusceptive angiogenesis (reviewed by Du Cheyne [20]), but very few of them relied on live imaging (and thus tracking one particular structure over a longer time), and/or provide 3D data to distinguish a pillar from an intravascular fold or a mesh. The novelty of our pilot experiments is in the methodology that enabled us to unambiguously interpret the imaging data. However, further investigations in different animal models are needed, as the pillar formation, and its further fate might also depend on the chosen model. Chicken chorioallantoic membrane, mouse retina, and capillary networks in the developing lung or kidney or in solid tumours might expand “within themselves”, whereas the CVP of zebrafish rather expands its borders. Alternatively, the behaviour of pillars in the CVP might be specific for lower vertebrates—which is, in our opinion, rather unlikely, since the homology between fish and mammalian genes is generally high. All this would require further investigation.

To summarise, this study presents, to our knowledge, the first time-lapse 3D images of pillar formation and development in vivo, in healthy animals, in a very high resolution. Our data demonstrated that, when discriminating IA from SA, it is important to distinguish between the thin intussusceptive pillars and the bigger intercapillary meshes. Firstly, pillars and small meshes differ in their shape and their content. Secondly, according to our findings, pillars and small meshes have different mechanisms of formation, the latter originating by sprouting angiogenesis and not by the growth of small pillars. In contrast, the formation of pillars seemed to be driven by three major processes, which we described earlier. Thirdly, they behave differently—possibly since they have a different content. This also means that they have different roles or a different fate in the developing plexus. Last but not least, we identified the absence of blood flow as a common positive predictor of high pillar density—in the *sih* model as well as after pharmacological intervention (LatB, BHG712). This knowledge could help to bring more light into the mechanisms of splitting angiogenesis and bridge the gap in the knowledge on SA-IA interplay.

## 4. Materials and Methods

### 4.1. Animals

Zebrafish (Danio rerio) were originally obtained from the Zebrafish International Resource Center, Eugene, OR, USA (https://zebrafish.org/home/guide.php, accessed on 3 January 2016). They were maintained at standard conditions in a conventional fish facility at the Institute of Anatomy of the University of Bern. Embryos were raised in an incubator in Petri dishes with an E3 medium at 28.5 °C and staged by hpf. The following zebrafish lines were used in this study: *Tg(fli1a :eGFP)y7* [61], *Tg(gata1: DsRed)sd2* [62], and *sih* (bearing a mutation in *tnnt2a*, a gene encoding cardiac troponin T) [63].

For live microscopy, embryos were manually dechorionated at 24 hpf and mounted from 28 hpf in glass-bottom Petri dishes in an E3 medium containing 0.167% low-melting agarose, 0.0175% tricaine, and 72 μM phenylthiourea (all from Sigma-Aldrich, St. Louis, MI, USA) [27].

The following selected antiangiogenic substances were used: Latrunculin B (LatB): 250 nM (Sigma-Aldrich, St. Louis, MI, USA), Sorafenib: 1 µM (Selleckchem, Houston, TX, USA), cediranib: 5 µM (Selleckchem, Houston, TX, USA), and BHG712: 1 µM (Sigma-Aldrich, St. Louis, MI, USA). The stock solutions were prepared according to the manufacturer’s instructions, and the reagents were diluted to a desired concentration in a fish medium E3. We started the treatment at 24 hpf and imaged the embryos at ca 48 hpf. Pillar count per plexus was from at least 4 replicates, and we further acquired at least two time-lapse scans pro inhibitor from ca 32 hpf until ca 48 hpf using the same methodology as for the untreated control embryos (LSM880, 40x/1.1 obj., see details in the next section).

### 4.2. Live Imaging

Confocal scans of the developing CVP were acquired using a Zeiss (Jena, Germany) LSM880 microscope, objective: 40x/1.1 LD-C-Apochromat W (https://www.zeiss.com, accessed on 3 January 2016), in a heated and humidified chamber (28.5 °C) with a resolution of 0.16–0.35 μm in xy, 0.5–0.75 μm in z, and a time step of 10 min. The time point format hh:mm refers to the age of the embryo in hours and minutes post fertilisation. Mounting and prolonged imaging (10 h+) resulted in small deformations of the tail and a slight reduction in the body size and blood velocity (compare [27]), but no other signs of distress and no phototoxicity were observed, and the patterns of the CVP were indistinguishable from freely hatched siblings. Analyses of pillar morphology, pillar formation, development, mesh dynamics, and the quantification of the area change are based on 6 confocal scans starting between 28 and 32 hpf and ending at 44–48 hpf. Pillar count per plexus was based on 11–28 replicates per time point.

### 4.3. Serial Block Face Scanning Electron Microscopy

Forty-four hpf embryos were fixed with Karnovsky fixative (2% paraformaldehyde; 2% glutaraldehyde in 0.15 M Na-cacodylate buffer). Samples were rinsed 3 × 5 min in 0.15 M Na-cacodylate. They were then incubated in 2% OsO4 and 1.5% potassium ferrocyanide in 0.3 M Na-cacodylate for 1 h at room temperature. They were rinsed 3 × 5 min in water. They were incubated with 0.64 M pyrogallol for 20 min at room temperature and subsequently rinsed 3 × 5 min with water. The samples were incubated in 2% OsO4 for 30 min at room temperature. After 3 × 5 min rinses in water, the samples were incubated overnight in a solution of 0.15 M gadolinium acetate (LFG Distribution, Lyon, France) and 0.15 M samarium acetate (LFG Distribution) pH 7.0. They were then rinsed 3 × 5 min with water, incubated in 1% Walton’s lead aspartate [74] at 60 °C for 30 min, and rinsed 3 × 5 min with water.

After staining, the samples were dehydrated in a graded ethanol series (20%, 50%, 70%, 90%, 100%, and 100%) at 4 °C, each step lasting 5 min. They were then infiltrated with Durcupan resin mixed with ethanol at ratios of 1:3 (*v*/*v*), 1:1, and 3:1, each step lasting 2 h, and with pure Durcupan overnight. The samples were transferred to fresh Durcupan, and the resin was polymerised for 3 days at 60 °C. Sample blocks were mounted on aluminium pins (Gatan, Pleasonton, CA, USA) with conductive epoxy glue (CW2400, Circuitworks, Kennesaw, GA, USA). Care was taken to have osmicated material directly exposed at the block surface in contact with the glue in order to reduce specimen charging under the electron beam. Pyramids with a surface of approximately 500 × 500 μm^2^ were trimmed with a razor blade.

Three-dimensional (3D) ultrastructural images were produced via serial block face scanning electron microscopy (SBF SEM) on a Quanta FEG 250 SEM (FEI, Eindhoven, The Netherlands) equipped with a 3View2XP in situ ultramicrotome (Gatan). Image acquisition was carried out with a backscattered electron detector optimised for SBF SEM (Gatan). Images were taken with formatted voltage 5 kV, spot size 3000, and pixel time 0.5 µs. Z-stacks were acquired with pixel resolution of 12 nm and 150 nm z-step.4.4 Inhibitor treatment.

### 4.4. Pillar Tracking

A sub-volume of the total image size between 50 × 50 × 17 μm and 100 × 100 × 26 μm around each pillar was tracked in 3D over time on the full-size time-lapse image dataset, using software developed by Vladymyrov [75]. This software was initially designed to perform the correction of tissue drift in real time during in vivo acquisitions [30]. Its core is an algorithm performing fine pattern matching in 3D, which can be applied to a number of tracking tasks. A function to perform tracking of 3D sub-volumes was developed especially for this project (author: Mykhailo Vladymyrov/Data science lab, University of Bern, Bern, Switzerland, version: 21.9.2016, website: https://www.dsl.unibe.ch/about/contact_and_team/ accessed on 15 November 2023). In addition, since the tissue shape changed quickly during the development of the embryo, the alignment timeframe had to be reduced from 30 timeframes to 1 timeframe. This allowed for keeping high correlation values in the offsets probability map and enabled robust tracking of the region of interest. The tracking procedure output was a list of the pillar coordinates at each time point in the original dataset and a set of 3D image crops around this position. Those image crops were used for further visual inspection. Since the xyz coordinates of the pillars shifted as the plexus grew, the scans had to be processed so that the pillars were centred in a defined region of interest at each time point.

### 4.5. Image Analysis, Statistics, and Artwork

Measurements and analyses were conducted manually using Fiji (https://imagej.net/software/fiji/downloads accessed on 15 November 2023); the diameter and cross-section areas were measured at 44–48 hpf in the middle of the pillar; the accuracy of the length/height measurements was 0.5 µm, and the cross-section areas were rounded to full µm^2^. Three-dimensional representations of the endothelial surfaces were created in Imaris (version 9.9 or 8.X, Bitplane, Schlieren, Switzerland). Graphs were generated using Stata (also used for statistics) (Stata 15, Stata Corp. Ltd., College Station, TX, USA or Excel software (Excel 2019, Microsoft corp., Redmont, WA, USA) the illustrations were created in Inkscape (Version 0.93.2, community developed software, https://inkscape.org/de/ accessed on 15 November 2023 and MediBang Paint Pro (Version 26.2., MediBang Ltd., Tokyo, Japan). Data such as pillar length, height, and pillar count per plexus per time point are given as mean ± standard deviation.

## 5. Conclusions

We brought evidence that (a) transluminal pillars in the CVP formed via different mechanisms than the classical intussusception, (b) their formation was independent of the blood flow, (c) the thin pillars did not become thicker with time as expected but rather stayed the same or even shrank and disappeared, and (d) plexus remodelling was driven by the dynamics of bigger intercapillary meshes, not pillars. Further experiments are needed to decipher the pillar formation in mammalian vascular beds and to investigate whether they contribute to plexus remodelling. We showed that clear, unambiguous criteria are needed to discriminate between IA and SA, and we strongly suggest that evidence is based on time-lapse 3D data.

## Figures and Tables

**Figure 1 ijms-24-16703-f001:**
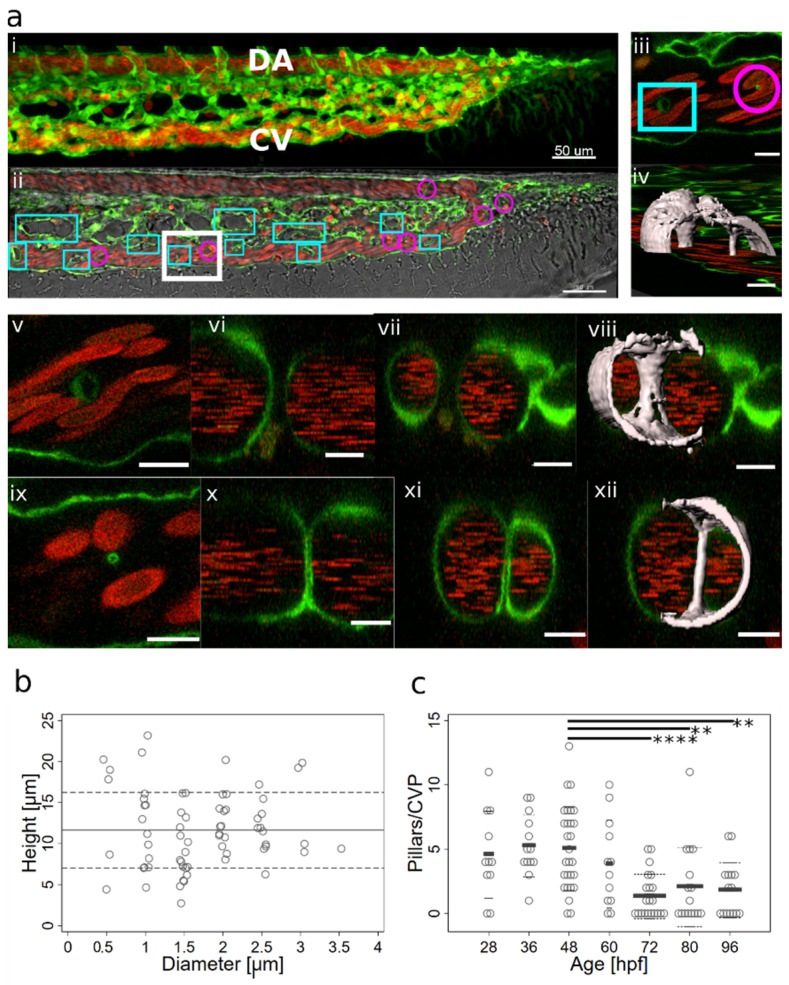
Morphology of pillars and meshes and pillar count per plexus. (**a**) Confocal images of Fli1a:eGFP//Gata1:dsRed embryos; endothelium in green (eGFP); erythrocytes in red (dsRed); and the wide field channel in greytone. (**i**,**ii**): Overview of the CVP at 48 hpf shows the blood flow in the dorsal aorta and the caudal vein (DA, CV); the tail of the fish is pointing to the right of the image. (**i**) is a maximum intensity projection of Z-stacks; (**ii**) a single slice image. Cyan rectangles show meshes, magenta circles, and pillars within the plexus; the region indicated by a white rectangle is shown in detail in (**iii**) (single slice) and (**iv**) (3D reconstruction of the endothelium based on the confocal data). The orientation of the pillar is perpendicular to the blood flow (moving erythrocytes seen as red streaks). (**v**,**vi**,**vii**,**viii**) show further details of the small mesh from image (**iii**); in (**v**) a cross-section (xy); in (**vi**,**vii**), a longitudinal section (yz and xz; perivascular cells infiltrating the mesh are apparent); and in (**viii**), the 3D reconstruction in grey. In (**ix**,**x**,**xi**,**xii**), details of the pillar from image (**iii**) are seen; in (**ix**, a cross-section (xy), in (**x**,**xi**), a longitudinal section (yz and xz); and in (**xii**), the 3D reconstruction in grey. Note the different shapes and diameters of the mesh in comparison to the pillar. Scale bar (**i**,**ii**): 50 µm; other images: 10 µm. (**b**) Distribution of pillar height (length in z-axis) and diameter (in xy plane) at 48 hpf. The thick line represents the average height; dashed lines represent the average plus or minus standard deviation. (**c**) Number of pillars per plexus measured in in vivo confocal images between 28 and 96 hpf (1 and 4 dpf). Thick lines represent the average pillar number for each time point, and thin dashed lines are the average plus or minus standard deviation. Significant differences between the different time points are marked with ** for a *p*-value ≤ 0.01 and **** for a *p*-value ≤ 0.0001.

**Figure 5 ijms-24-16703-f005:**
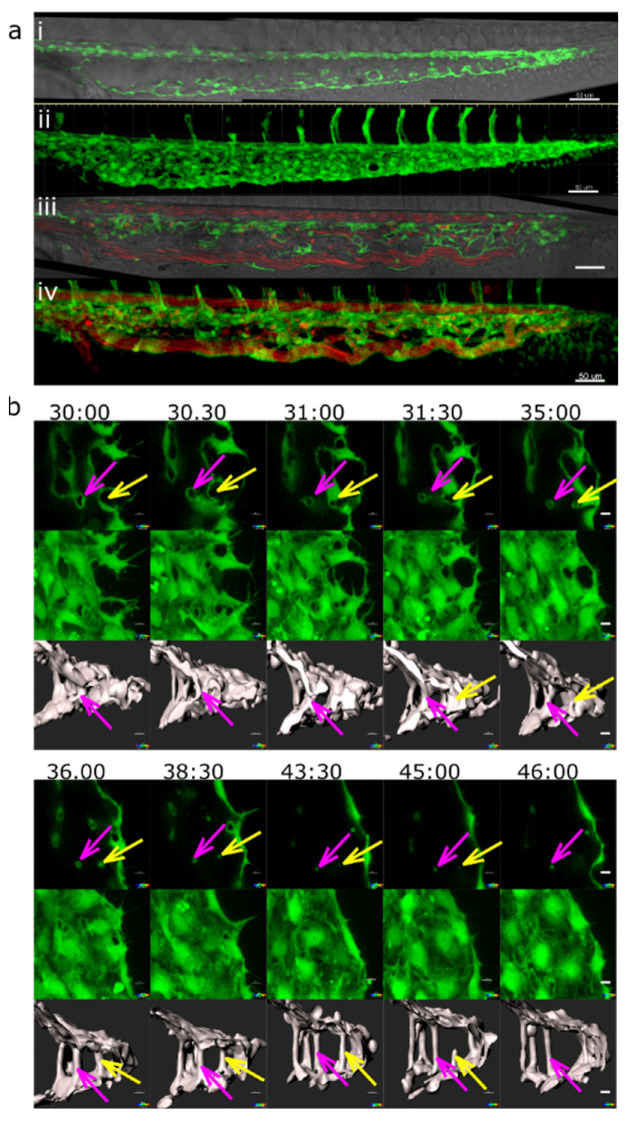
Pillar and mesh formation in the absence of blood flow. (**a**): CVP of a Fli:eGFP//*sih* embryo (**i**,**ii**) compared to the CVP of a wt embryo (**iii**,**iv**) at 48 hpf. The *sih* embryo does not have any blood flow and thus no local shear stress differences. I and III: single confocal section; (**ii**,**iv**): maximum intensity projection. Endothelium in green (eGFP), erythrocytes in red (dsRed), and bright field in grey (**i**,**iii**). Note that there are frequent pillars (visible as tiny circles in single confocal sections) but very few meshes (oval-like shapes corresponding to dark spots in (**ii**)). This stands in contrast to the wt CVP seen in (**iii**,**iv**), where there are only a few pillars visible. Scale bars: 50 µm. (**b**): Mechanism of pillar formation in the absence of blood flow. The driving forces of pillar formation in Fli:eGFP//*sih* embryos are sprouting and lumen expansion, i.e., the same phenomena as in wt embryos. The development of two pillars is marked by coloured arrows; one of them disappears at the end of the image sequence (time point 45:00–46:00; yellow arrow) while another pillar (magenta arrow) persists. First row: one confocal section. Second row: maximum intensity projection of the same region. Third row: surface models of the endothelium in grey (based on the GFP channel). Time points (hh:mm post fertilisation) are valid for each column. Scale bars: 5 µm.

**Figure 6 ijms-24-16703-f006:**
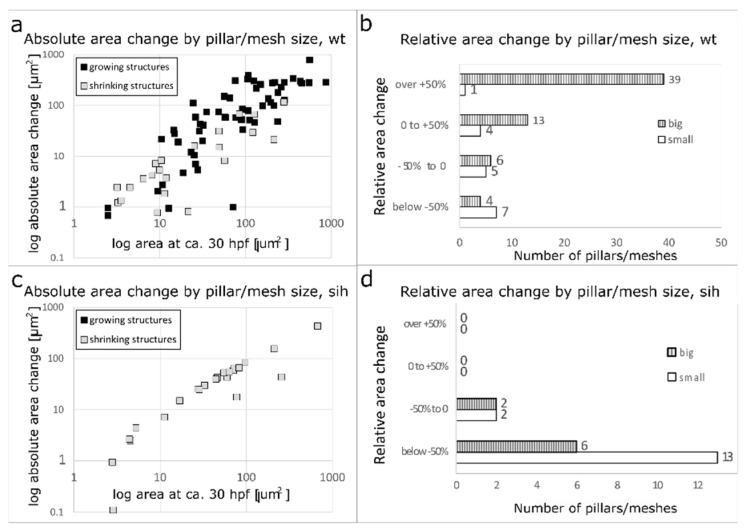
Absolute and relative changes in mesh/pillar area depending on their initial size and the presence or absence of blood flow. (**a**,**b**): wt embryos. (**c**,**d**): *sih* embryos (no blood circulation). Changes in pillar/mesh size at 44–48 hpf are plotted against their initial area at 28–32 hpf. In wt (**a**), the bigger the mesh, the more it usually grew or shrank, while in *sih* (**c**), all measured meshes shrank independently from their initial size. Relative changes in pillar/mesh area are seen in (**b**,**d**): In wt, very small meshes and pillars with an area <12 µm^2^ at 28–32 hpf (corresponding to a diameter below 4 µm, white bars in the tables) did not grow or shrink much, while the majority of bigger meshes (>12 µm^2^, striped bars) changed their size (**b**). In *sih*, all the structures shrank, independently from their initial size (**d**).

**Figure 7 ijms-24-16703-f007:**
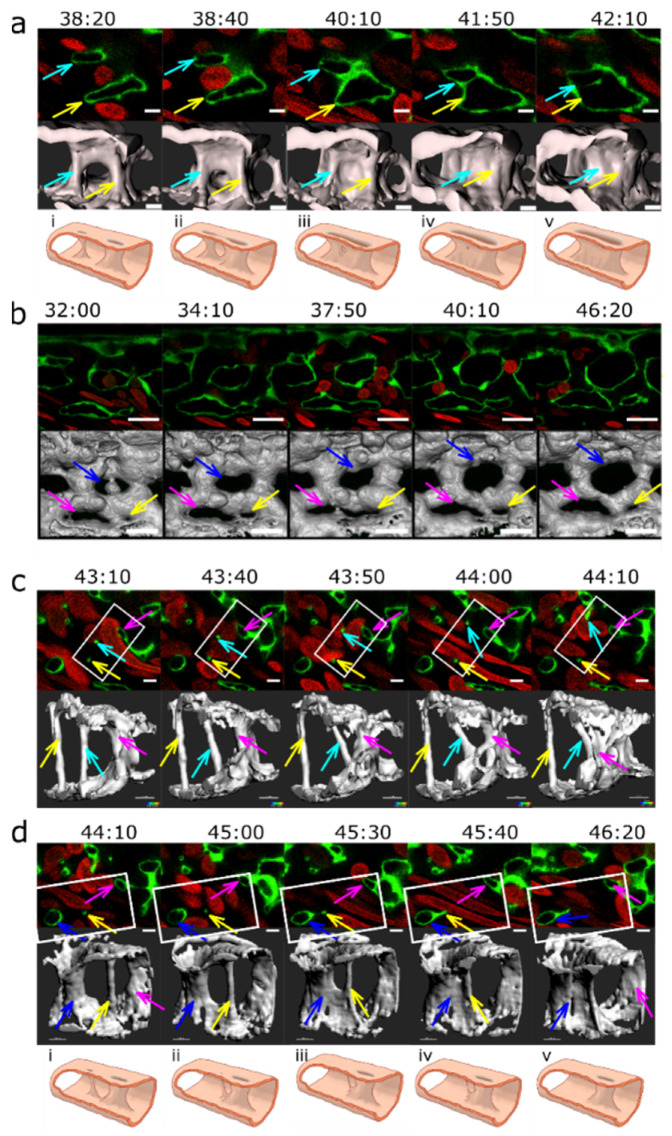
Vascular splitting by merging of meshes; pillar disappearance by merging with a mesh (capillary wall folding reversed). Different examples of pillar and mesh behaviour seen in confocal images. Endothelium in green (eGFP); erythrocytes in red (dsRed). First row: single slice. Second row: 3D reconstruction of the endothelium in grey (based on the GFP channel). (**a**) Example of merging of two small meshes, marked by arrows (yellow and light blue). Scale bar: 5 µm. (**i**–**v**): illustration of the proposed mechanism based on the 3D reconstructions above. (**b**) Example of a growing mesh (blue arrow) and a reversible mesh merging (yellow and magenta arrows). Note the increasing diameter of the mesh marked with the blue arrow between time point 37:50 and 40:10 and again at 46:20. The two meshes marked with a yellow and a magenta arrow fuse at time point 37:50 and separate again at 40:10. Scale bars: 20 µm. (**c**,**d**) Disappearance of several pillars in one region by merging with meshes. The pillar marked with a yellow arrow fuses with the mesh on the bottom left of the image (blue arrow) between time point 44:10 and 46:20. The other pillar, marked with a light blue arrow, fuses with another mesh between time point 43:40 and 44:10. Unlike merging of meshes, the fusion of a pillar with a nearby structure is not accompanied by a visible increase in its size. The 3D reconstruction was conducted only from the area within the white rectangle. Time points (hh:mm post fertilisation) are valid for each column. Scale bar: 5 µm. (**i**–**v**): illustration of the proposed mechanism of the 3D reconstructions above.

**Figure 8 ijms-24-16703-f008:**
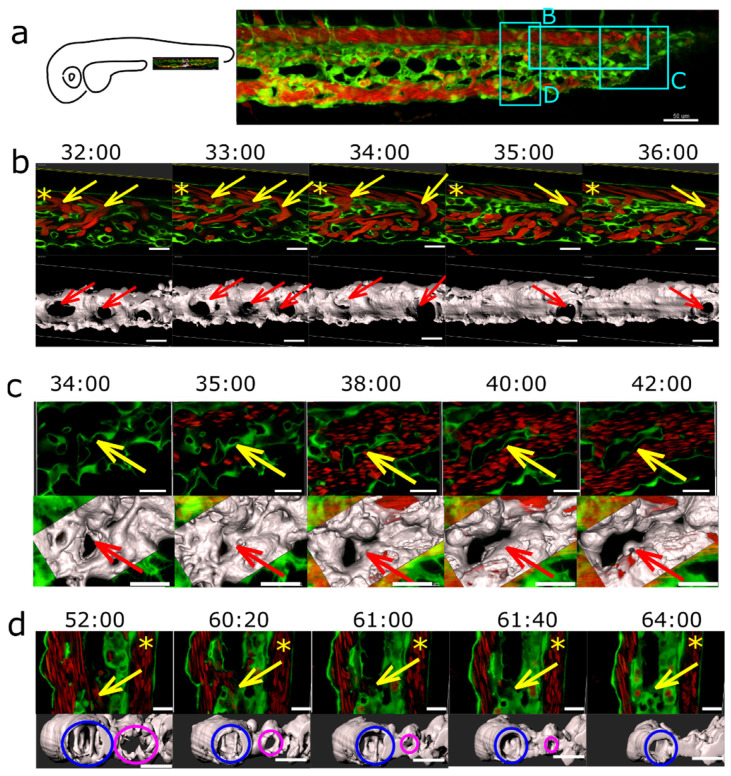
Arteriovenous and venous splitting. (**a**): Overview of the CVP of Fli1a:eGFP//Gata1:dsRed embryos showing the regions (rectangles B–D) where image sequences (**b**–**d**, in the same order) were taken (since these were from different embryos, this is just a typical example CVP). DA is in (**b**,**d**) marked with an asterisk. Endothelium in green, erythrocytes in red, and 3D reconstruction in grey (based on the eGFP channel). (**b**): Arteriovenous splitting. The image sequence shows connections between the DA and the CV marked with yellow arrows that gradually close except for the most caudal one; thus, the blood flow shifts towards the caudal end of the plexus. First row: single slice. Second row: 3D reconstruction, showing the arteriovenous connections (red arrows). (**c**): Venous splitting at the growing front of the plexus occurs by merging of meshes (yellow arrow or for better visibility red arrow on the 3D reconstruction). First row: single slice. Second row: 3D reconstruction within the rectangle in grey. (**d**): (Intussusceptive) Arborisation: The lumen of a redundant venous segment (yellow arrow or for better visibility magenta circle on the 3D reconstruction) in the central part of the CVP gradually shrinks and eventually fully collapses, leaving only the lumen of the caudal vein (blue circle) patent. First row: single slice. Second row: 3D reconstruction in grey. Time points (hh:mm post fertilisation) are valid for both columns, respectively. Scale bars: in (**a**): 50 µm; (**b**–**d**): 20 µm.

**Figure 9 ijms-24-16703-f009:**
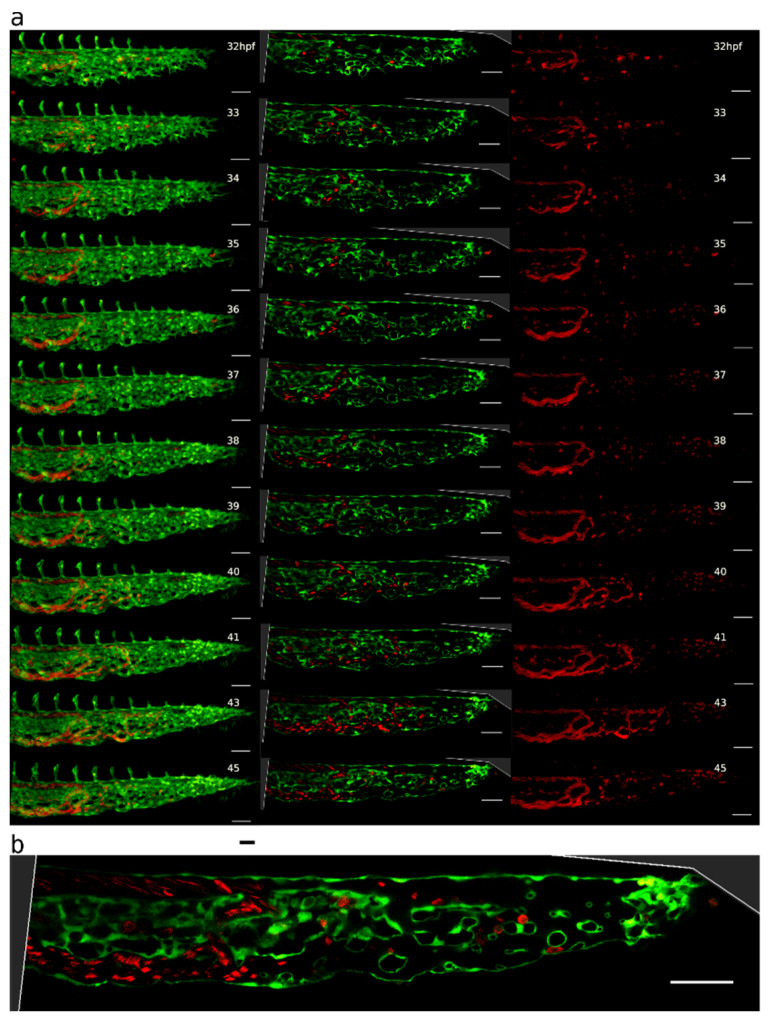
Inhibition of arteriovenous splitting with 1 µM BHG712 on a *Fli1a:eGFP//Gata1:dsRed* model. (**a**) Confocal time-lapse scan of *Fli1a:eGFP//Gata1:dsRed* embryos; endothelium in green (eGFP); erythrocytes in red (dsRed). First column: maximum intensity projection of Z-stacks; second column: single slice images/Time-lapse scan; first column: maximum intensity projection; second column: single section; third column: only red channel (Gata:dsRed expressing erythrocytes) showing the blood flow. (**b**) Higher magnification of a single confocal section. Note that the distal part of CVP has lumen but is not perfused; it only contains only isolated erythrocytes. Note the numerous small meshes and pillars present (seen as small circles in cross-section). Scale bars: 50 µm.

## Data Availability

Data are contained within the article and Appendix A.

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
