# Peer review of "Transluminal Pillars—Their Origin and Role in the Remodelling of the Zebrafish Caudal Vein Plexus"

_ijms, 2023, doi:10.3390/ijms242316703_

Round 1

Reviewer 1 Report

Comments and Suggestions for Authors

The manuscript titled “transluminal Pillars - their Origin and Role in the Remodelling of the Zebrafish Caudal Vein Plexus” has been well written but the way it is structured now it doesn’t bring out a strong standout point that can make it publishable so therefore below are my concerns that needs to be addressed.

Minor concerns

1.     In Fig 6 b and d can authors also show total average of pillars between wt and sih

2.     As it is stated in the first paragraph of discussion it is hard to make out the novel finding of this manuscript, I think there is need to rewrite some important aspects and highlighting the key observation of manuscript more precisely. 

3.     It is important to explain the physiological importance of various mechanism of vessel splitting. 

4.     For all the figures it is important to write what objectives were used at confocal microscope

5.     In sih model if pillars seem to be not affected than what is the phenotype of mutant

6.     The sentence” in line with that, we observed numerous transluminal pillars in the CVP during that time” doesn’t refer to particular figure, please mentioned that

Major concern 

1.     Functional relevance of pillars and IA

2.     The pillars that authors see in Fig 2, 3 and 4 with confocal microscopy, have they figured out the molecular composition of pillars and how it differs from meshes.

3.     Did authors try to create physical injury model and try to compare vessel development and see if is similar to sih model ?

4.     It will be important to show labelling that are specific of pillars and mesh that can help in differentiating the two processes from each other.

Comments on the Quality of English Language

Overall English is fine, scientific explanation needs a clear approach in the way it is written.

Author Response

Lieber Rezensent

Unsere Antworten auf Ihre wertvollen Kommentare finden Sie im beigefügten Dokument.

Reviewer 2 Report

Comments and Suggestions for Authors

Ross and colleagues, through beautiful time-lapse imaging described pillar/mesh formation in the CVP in zebrafish. The imaging data is convincing and clearly displayed. However, mechanistic insight controlling pillar/mesh formation is not explored in detail, making it a highly descriptive finding based mainly on timelapse video between 30-48 hpf. Considering sprouting angiogenesis is required for both pillar and mesh formation, it will be important to explore roles of the Vegfr pathway. Below suggestions to improve manuscript.

Major

As a major driver for sprouting angiogenesis, It would be essential to determine if Vegfa/Vegfr2 signalling is required for the pillar formation. A Vegfr inhibitor (eg SU5416) could be treated from 30 hpf to determine if the pillar/mesh number is modulated. This would provide some mechanistic insight to this newly characterised structure.

Is there an increase/decrease in pillar/mesh number in sih mutants? Please quantify pillar number in mutants and wild type siblings with flow.

Figures 2-4,5,7,8 – Please provide annotated timelapse movies of the representative images.

Minor:

Line 135 – Please provide a supplementary image of the pillar located at the dorsal aorta.

Figure 1c – Please provide p value demonstrating by significance between 36-48hpf and 3 dpf onwards.

Is the supplementary images not referenced in the main text?

Author Response

Dear reviewer

Please find our answers to your valuable comments on our manuscript in the attached document.

Reviewer 3 Report

Comments and Suggestions for Authors

The research article by Röss et al., entitled “Transluminal Pillars - their Origin and Role in the Remodelling of the Zebrafish Caudal Vein Plexus” investigates, pillar formation and their further maturation by in vivo confocal microscopy in high spatio-temporal resolution using the transgenic zebrafish model Fli1a:eGPF//Gata1:dsRed and by employing zebrafish caudal vein plexus (CVP) in intussusceptive angiogenesis. The article is interesting and may be useful for understanding the Intussusceptive angiogenesis, a dynamic intravascular process capable of dramatically modifying the structure of the microcirculation.

I have following Minor concerns about the article.

1) The zebrafish lines used in this study: Tg(fli1a :eGFP)y7, Tg(gata1: DsRed)sd2, and sih need a brief description in the introduction section of the article because most of the results presented in the study are based on these zebrafish lines, this will make the article easier to read, understand and interpret the results.

2) In the legend of figure 6, it should be µm2, not the µm2. On the page 9, line number 279 and 283, authors have written, the average relative cross-section area change was -16±68% (absolute average change +1.1±9.9 μm2 and the average relative difference in the cross-section area of meshes between the two time points was +95±110% (absolute average change +112±150 μm2), for both of these results, SD are very high as compared to average value, authors need to explain this in the results section.

3) Page 5, line number 194, authors have written, were very unstable, got thinner und broke within an hour or two. It should be and not und.

Author Response

Dear reviewer

Please find our answers to your valid comments on our manuscript in the attached document.

Round 2

Reviewer 1 Report

Comments and Suggestions for Authors

Authors have addressed all my concerns appropriately, I am convinced manuscript should be helpful for the scientific community towards increasing their knowledge.

Reviewer 2 Report

Comments and Suggestions for Authors

The authors have addressed all my concerns and I congratulate the authors for this very beautiful paper.